# Neuromorphologicaly-preserving Volumetric data encoding using VQ-VAE

| | |
|---|---|
| *******[1] | **@** |
| *******[1] | **@** |
| *******[1,2] | **@** |
| *******[1,2] | **@** |
| *******[1,2] | **@** |
| *******[1] | **@** |
| *******[3] | **@** |
| *******[1] | **@** |
| *******[1] | **@** |

## Abstract

The increasing efficiency and compactness of deep learning architectures, together with hardware improvements, have enabled the complex and high-dimensional modelling of medical volumetric data at higher resolutions. Recently, Vector-Quantised Variational Autoencoders (VQ-VAE) have been proposed as an efficient generative unsupervised learning approach that can encode images to a small percentage of their initial size, while preserving their decoded fidelity. Here, we show a VQ-VAE inspired network can efficiently encode a full-resolution 3D brain volume, compressing the data to 0.825% of the original size while maintaining image fidelity, and significantly outperforming the previous state-of-the-art. We then demonstrate that VQ-VAE decoded images preserve the morphological characteristics of the original data through voxel-based morphology and segmentation experiments. Lastly, we show that such models can be pre-trained and then fine-tuned on different datasets without the introduction of bias.

**Keywords:** 3D, MRI, Morphology, Encoding, VQ-VAE

## 1. Introduction

It is well known that Convolutional Neural Networks (CNN) excel at a myriad of computer vision (CV) tasks such as segmentation (Ronneberger et al., 2015; Kohl et al., 2018), depth estimation (Moniz et al., 2018) and classification (Xie et al., 2019). Such success stems from a combination of dataset size, improved compute capabilities, and associated software, making it ideal for tackling problems with 2D input imaging data. Medical data, however, has limited data availability due to privacy and cost, and causes increased model complexity due to the volumetric nature of the information, making its modelling non-trivial. Unsupervised generative models of 2D imaging data have recently shown excellent results on classical computer vision datasets (van den Oord et al., 2017; Brock et al., 2018; Razavi et al., 2019; Ho et al., 2019; Donahue and Simonyan, 2019), with some promising early results on 2D medical imaging data (Bass et al., 2019; Gupta et al., 2019; Lee et al., 2019; Liu et al., 2019). However, approaches on 3D medical data have shown limited sample fidelity

(Kwon et al., 2019; Choi et al., 2018; Zhuang et al., 2019), and have yet to demonstrate that they are morphology preserving.

From an architectural and modelling point of view, Generative Adversarial Networks (GAN) (Goodfellow et al., 2014) are known to have a wide range of caveats that hinder both their training and reproducibility. Convergence issues caused by problematic generator-discriminator interactions, mode collapse that results in a very limited variety of samples, and vanishing gradients due to non-optimal discriminator performance are some of the known problems with this technique. Variational Autoencoders (VAE) (Kingma and Welling, 2013), on the other hand, can mitigate some convergence issues but are known to have problems reconstructing high frequency features, thus resulting in low fidelity samples. The Vector Quantised-VAE (VQ-VAE) (van den Oord et al., 2017) was introduced by Oord *et al.* with the aim of improving VAE sample fidelity while avoiding the mode collapse and convergence issues of GANs. VQ-VAEs replace the VAE Gaussian prior $\mathcal{N}(0, 1)$ with a vector quantization procedure that limits the dimensionality of the encoding to the amount of atomic elements in a dictionary, which is learned either via gradient propagation or an exponential moving average (EMA). Due to the lack of an explicit Gaussian prior, sampling of VQ-VAEs can be achieved through the use of PixelCNNs (Salimans et al., 2017) on the dictionary's elements.

In this work we propose a 3D VQ-VAE-inspired model that successfully reconstructs high-fidelity, full-resolution, and neuro-morphologically correct brain images. We adapt the VQ-VAE to 3D inputs and introduce SubPixel Deconvolutions (Shi et al., 2016) to address grid-like reconstruction artifacts. The network is trained using FixUp blocks (Zhang et al., 2019) allowing us to stably train the network without batch normalization issues caused by small batch sizes. We also test two losses, one inspired by (Baur et al., 2019) and a 3D adaptation of (Barron, 2019). Lastly, to demonstrate that decoded samples are morphologically preserved, we run VBM analysis on a control-vs-Alzheimer's disease (AD) task using both original and decoded data, and demonstrate high dice and volumetric similarities between segmentations of the original and decoded data.

## 2. Methods [1]

### 2.1. Model Architecture

The original VAE is composed of an encoder network that models a posterior $p(z|x)$ of the random variable $z$ given the input $x$, a posterior distribution $p(z)$ which is usually assumed to be $\mathcal{N}(0, 1)$ and a distribution $p(x|z)$ over the input data via a decoder network. A VQ-VAE replaces the VAE's posterior with an embedding space $e \in R^{K \times D}$ with $K$ being the number of atomic elements in the space and $D$ the dimension of each atomic element $e_i \in R^D, i \in 1, 2, \ldots, K$. After the encoder network projects an input $x$ to a latent representation $z_e(x)$ each feature depth vector (the features corresponding to each voxel) is quantized via nearest neighbour look-up through the shared embedding space $e$. The posterior distribution can be seen as a categorical one defined as follows:

$$q(z = k|x) = \begin{cases} 1 & \text{for k} = \text{argmin}_j \|z_e(x) - e_j\|_2 \\ 0 & \text{otherwise} \end{cases} \quad (1)$$

---

1. The code will be available at the time of publication on GitHub

This can be seen through the prism of a VAE that has an ELBO of $\log p(x)$ and a KL divergence equal to $\log K$ given that the prior distribution is a categorical one with $K$ atomic elements. The embedding space is learned via back propagation or EMA (exponential moving average).

We modified the VQ-VAE to work with 3D data as shown in Figure 1. Firstly, all 2D convolutional blocks were replaced with 3D blocks due to the 3D nature of the input data and information that we want to encode. To limit the dimensionality of the model and of the representation, the VQ blocks were only introduced at $48^3$, $12^3$ and $3^3$ resolutions. The number of features was doubled after each strided convolution block starting from the maximum number of features that would fit a GPU at full resolution. Our residual blocks are based on the FixUp initialization (Zhang et al., 2019) so we circumvent any possible interference from the batch statistics being too noisy due to memory constraints. Furthermore, we are using transpose convolutions with a kernel of 4 and ICNR initialization (Aitken et al., 2017) followed by an average pooling layer with a kernel of 2 and stride of 1 (Sugawara et al., 2019) for upsampling the activations. The last upsampling layer uses a subpixel convolution (Shi et al., 2016; Aitken et al., 2017; Sugawara et al., 2019) in order to counter the checkerboard artifacts that the transpose can generate.

The current architecture means that input data is compressed to a representation that is only 3.3% of the original image in terms of number of variables. More specifically, they are composed of three levels, the top one is $48\times64\times48\times2$, the middle one is $12\times16\times12\times8$ and the bottom one is $3\times4\times3\times32$. Note, however, that the quantized parameterization is encoded as a 8-bit integer while the original input as a 32-bit float, making the bit-wise compression rate 0.825% of the original size. The higher resolution codes are conditioned on the immediately lower resolution one which encourages them not to learn the same information.

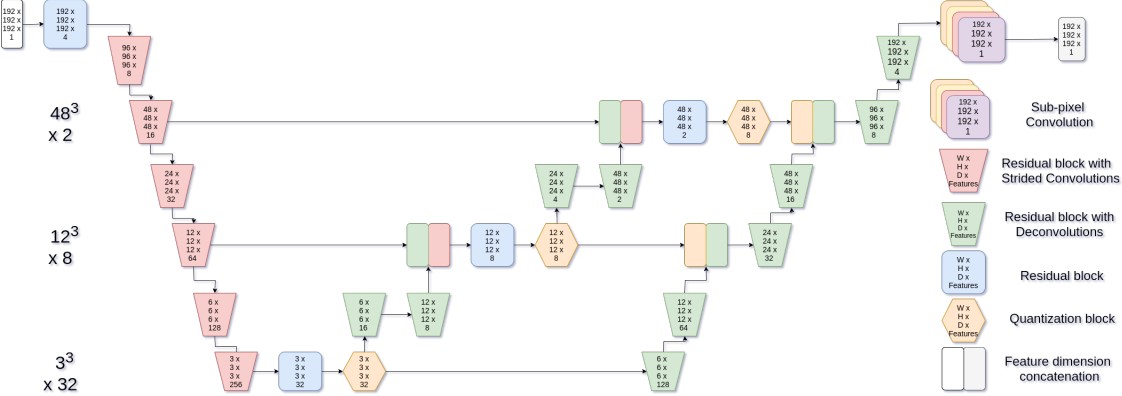

Figure 1: Network architecture - 3D VQ-VAE

## 2.2. Loss Functions

For the image reconstruction loss, we propose to use the loss function from (Baur et al., 2019). The mathematical formulation is:

$$\mathcal{L}_{baur}(\mathbf{x}, \hat{\mathbf{x}}) = ||\mathbf{x} - \hat{\mathbf{x}}||_1 + ||\mathbf{x} - \hat{\mathbf{x}}||_2 + ||\nabla\mathbf{x} - \nabla\hat{\mathbf{x}}||_1 + ||\nabla\mathbf{x} - \hat{\nabla}\mathbf{x}||_2 \tag{2}$$

Due to the non-ideal nature of the L1 and L2 losses from a fidelity point of view, we explore the use of an adaptive loss as decribed in (Barron, 2019). We extend this adaptive loss to work on single-channel 3D volumes as input rather than 2D images. This loss automatically adapts itself during training by learning an alpha and scale parameter for each output dimension so that it is able to smoothly vary between a family of loss functions (Cauchy, Geman-McClure, L1, L2, etc.).

$$\mathcal{L}_{adaptive}(x, \alpha, c) = \frac{|\alpha - 2|}{\alpha} \left( \left( \frac{(x/c)^2}{|\alpha - 2|} + 1 \right)^{\alpha/2} - 1 \right) \tag{3}$$

As demonstrated in the original paper, the best results using the adaptive loss rely on some image representation besides pixels, for example, the discrete cosine transform (DCT) or wavelet representation. To apply this to 3D MRI volumes, we take the voxel-wise reconstruction errors and compute 3D DCT decompositions of them, placing the adaptive loss on each of the output image dimensions. 3D DCTs are simple to compute as we can use the separable property of the transform to simplify the calculation by doing a normal 1D DCT on all three dimensions. This works much better than using it on the raw pixel representation, as the DCT representation will avoid some of the issues associated with requiring a perfectly aligned output space, and it will model gradients instead of pixel intensities which almost always works better. The codebook losses are exactly the ones used in the original VQ-VAE 2 paper (Razavi et al., 2019) and implemented within Sonnet (https://github.com/deepmind/sonnet).

$$\mathcal{L}(\mathbf{x}, \mathbf{e}) = \|sg[E(\mathbf{x})] - \mathbf{e}\|_2^2 + \beta \|sg[\mathbf{e}] - E(\mathbf{x})\|_2^2 \tag{4}$$

Where $sg$ refers to a stop-gradient operation that blocks gradients from flowing through $\mathbf{e}$. As per (van den Oord et al., 2017; Razavi et al., 2019) the second term of the codebook loss was replaced with an exponential moving average for faster convergence:

$$N_i^{(t)} := N_i^{(t-1)} * \gamma + n_i^{(t)}(1 - \gamma), m_i^{(t)} := m_i^{(t-1)} * \gamma + \sum_j^{n^{(i)}} E(x)_{i,j}^{(t)}(1 - \gamma), e_i^{(t)} := \frac{m_i^{(t)}}{N_i^{(t)}} \tag{5}$$

where $\gamma$ is the decay parameter, $e_i$ is the codebook element, $E(x)$ are the features to be quantized and $n_i^{(t)}$ is the number of vectors in the minibatch.

## 3. Dataset and Preprocessing

The networks are first trained on a dataset of T1 structural images labeled as controls from the ADNI 1,GO,2 (Petersen et al., 2010; Beckett et al., 2015), OASIS (LaMontagne et al., 2018) and Barcelona studies (Salvadó et al., 2019). We skull stripped the images by first generating a binary mask of the brain using GIF (Cardoso et al., 2015), then blurring the mask to guarantee a smooth transition from background to the brain area and then superimposing the original brain mask to guarantee that the brain is properly extracted. Following

that, we registered the extracted brain to MNI space. Due to the memory requirements of the baseline network (Kwon et al., 2019), images were also resampled to 3mm isotropic to test how different methods worked at this resolution. We set aside 10% for testing results, totaling 1581 subjects for the training dataset and 176 subjects for the testing dataset. The images have been robust min-max scaled and no spatial augmentations have been applied during the training as images were MNI aligned. For fine-tuning, we used the Alzheimer's Disease (AD) patients from the ADNI 1, GO, 2 (Petersen et al., 2010; Beckett et al., 2015) datasets. The preprocessing and data split is identical to the control subjects dataset. For training we have 1085 subjects, while for testing we have 121 subjects.

## 4. Experiments and Results

### 4.1. Model Training Details

Our models were run on NVIDIA Tesla V100 32GB GPUs. The networks were implemented using NiftyNet (Gibson et al., 2018). The chosen optimizer was Adam (Kingma and Ba, 2014) combined with the SDGR learning rate scheduler (Loshchilov and Hutter, 2017) and a starting learning rate of $1e^{-4}$. Depending on the combination of architecture and loss function, we set the batch size to the maximum allowed size given GPU memory (ranging from batch size 32 for Kwon et al, to 512 for the proposed method at low res, and 8 for the proposed method at full resolution). The Control Normal models have run for 7 days and the fine-tuning with pathological data was run for an additional 4 days. To have a fair comparison we also trained models on the pathological data from scratch for the same

| Network | Net Res | Met Res | Tr Mode | MS-SSIM | log(MMD) | Dice WM | Dice GM | Dice CSF |
|---------|---------|---------|---------|---------|----------|---------|---------|----------|
| $\alpha$-WGAN | Low | High | H | 0.496 | 15.676 | 0.77+-0.03 | 0.86+-0.01 | 0.68+-0.03 |
| VQ-VAE Baur | High | High | H | **0.998** | 6.737 | **0.85+-0.05** | 0.90+-0.03 | 0.75+-0.09 |
| VQ-VAE Adap | High | High | H | 0.991 | **6.655** | 0.84+-0.05 | **0.92+-0.02** | **0.79+-0.08** |
| $\alpha$-WGAN | Low | High | P | 0.510 | 15.801 | 0.76+-0.02 | 0.86+-0.01 | 0.73+-0.03 |
| VQ-VAE Baur | High | High | P | **0.995** | 7.508 | **0.88+-0.07** | 0.91+-0.06 | 0.81+-0.10 |
| VQ-VAE Adap | High | High | P | 0.981 | **7.346** | 0.86+-0.05 | **0.94+-0.02** | **0.86+-0.05** |
| $\alpha$-WGAN | Low | High | PB | 0.511 | 15.807 | 0.75+-0.02 | 0.85+-0.01 | 0.73+-0.03 |
| VQ-VAE Baur | High | High | PB | **0.993** | 10.818 | **0.88+-0.05** | 0.92+-0.07 | 0.82+-0.09 |
| VQ-VAE Adap | High | High | PB | 0.984 | **7.573** | **0.88+-0.05** | **0.93+-0.02** | **0.84+-0.06** |

Table 1: Subset of metrics comparison between models at High (192x256x192) resolution. The "Net Res" is the resolution of the image passed to the network and "Met Res" is the resolution at which the metrics are calculated. The full table is available in Appendix. In bold are the best performers on a per "Tr Mode" and metric basis. We are showing Maximum Mean Discrepancy on log scale for ease of comparison. For resolution *Low* means *64x64x64* and *High* is *192x256x192*. For training mode *H* is *control subjects*, *P* is a model *fine-tuned on pathological data* after being trained on H and *PB* is a model *trained from scratch only on pathological dataset*.

amount of time as the fine-tuning. The best baseline model that we found is (Kwon et al., 2019). The authors propose an VAE - alpha Wasserstein GAN with Gradient Penalty based approach and encode brain volumes of size $64 \times 64 \times 64$ to a one dimensional tensor of length 1000. In our experiments we compress images to the same extent. Both of our methods use the ADNI dataset (Beckett et al., 2015).

### 4.2. Results

Table 1 details the quantitative results on image reconstruction fidelity. To provide comparison with (Kwon et al., 2019) we have measured Maximum Mean Discrepancy (MMD) (Gretton et al., 2012) and Multi-Scale Structural Similarity (MS-SSIM) (Wang et al., 2003). Significant improvements in fidelity were observed with the proposed method, both at 3mm and full resolution. We measured Dice (Milletari et al., 2016) overlap between segmentations of Gray Mater (GM) and White Matter (WM) in the ground truth and reconstructed volumes, and Cerebrospinal Fluid (CSF) as a proxy to the neuromorphological correctness of the reconstructions. The segmentations were extracted from the unified normalisation and segmentation step of Voxel Based Morphometry (VBM) (Ashburner and Friston, 2000)

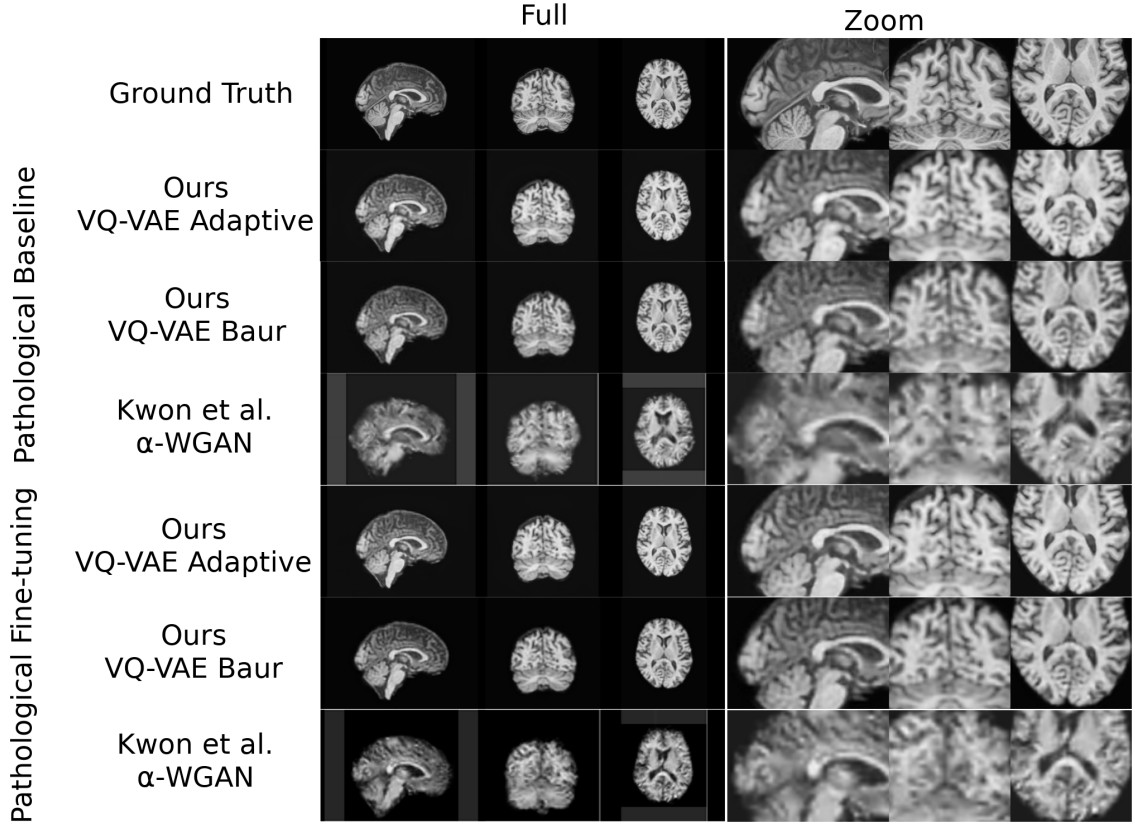

Figure 2: Comparison of a random reconstruction across the pathological training modes. $\alpha$-WGAN's reconstruction was upsamled to 1mm isotropic.

pipeline of Statistical Parametric Mapping (Penny et al., 2011) version 12. All metrics have been calculated over the test cases. Again, the proposed methods achieved statistically-significant ($p < 0.01$ Wilcoxon signed rank test) improved Dice scores against the $\alpha$-WGAN baseline, interchangeably between the Baur and Adaptive loss function.

## 5. Discussion

It can clearly be seen that our VQ-VAE model combined with the 3D Adaptive loss achieves the best performance in all three training modes. Interestingly the Baur loss trained model consistently performs better on MS-SSIM then the adaptive one. This might be attributed to the fact that the reconstructions of the adaptive appear like they have been passed through a total variation (TV) filter (Rudin et al., 1992) which could interfere with the fine texture of white matter. This indicates the need for future research in a hybrid loss function that is able to work better at a texture level, possibly combining the adaptive loss with image gradient losses. Even though the die scores are excellent which indicates possible good neuromorphometry, we would like to refer the reader to the VBM analysis that follows for a more in depth analysis since the SPM implementation is known to be highly robust.

VBM was performed to test for differences in morphology caused by the reconstruction process. Figure 3 displays the VBM analysis of the grey matter of the AD patient subgroup, comparing the original data and the reconstructed data. Ideally, if no morphological changes are observed, the t-map should be empty. Results show that the method by Kwon *et al.* show large t-values, while the proposed method shows significantly lower (closer to zero) maps, corroborating the hypothesis that the proposed method is more morphologically preserving.

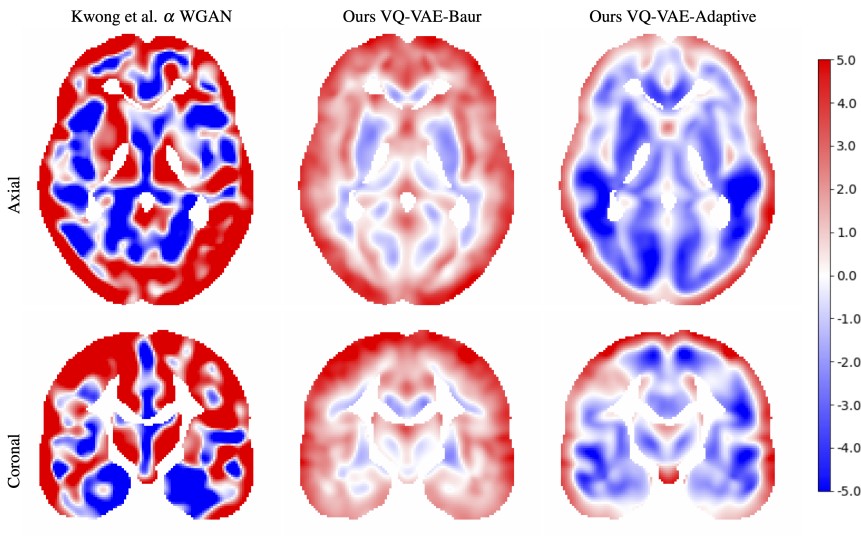

Figure 3: VBM Two-sided T-test between AD images and their reconstructions.

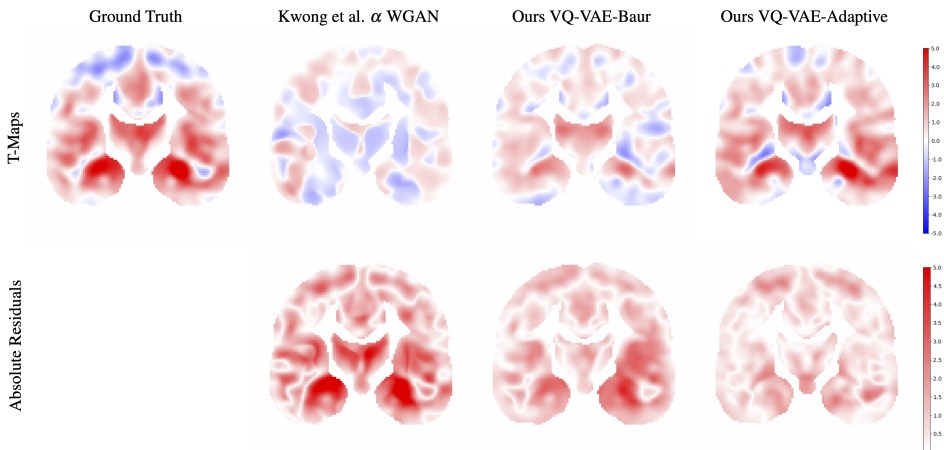

Figure 4: VBM Two-sided T-test between AD images and HC ones.

Lastly, in Figure 4, we looked at the T-test maps between AD and HC patients at the original resolution using the original data (labeled in the figure as ground truth) and then again using the reconstructed data for each method. In contrast to Figure 3, the best performing model is the VQ-VAE with Adaptive loss, where similar T-map clusters are observed, with low t-map residuals throughout. This means the proposed VQ-VAE Adaptive model was able to better learn the population statistics even though the MS-SSIM was found to be marginally lower when compared with the VQ-VAE with Baur loss. The discrepancy might be due to structural nature of the brain which is enforced by the sharper boundaries between the tissues of the TV filter like reconstructions in comparison with the more blurrier VQ-VAE Baur based output as seen in Figure 2.

## 6. Conclusion

In this paper, we introduced a novel vector-quantisation variational autoencoder architecture which is able to encode a full-resolution 3D brain MRI to a 0.825% of its original size whilst maintaining image fidelity and image structure. Higher multi-scale structural similarity index and lower maximum mean discrepancy showed that our proposed method outperformed the existing state-of-the-art in terms of image consistency metrics. We compared segmentations of white matter, grey matter and cerebro-spinal fluid in the original image and in the reconstructions showing improved performance. Additionally, VBM was employed to further study the morphological differences both within original and reconstructed pathological populations, and between pathological and control ones for each method. The results confirmed that both variants of our VQ-VAE method preserve the anatomical structure of the brain better than previously published GAN-based approaches when looking at healthy brains and those with Alzheimer's disease. We hope that this paper will encourage further advances in 3D reconstruction and generative 3D models of medical imaging.

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

## Appendix

| Network | Net Res | Met Res | Tr Mode | MS-SSIM | log(MMD) | Dice WM | Dice GM | Dice CSF |
|---|---|---|---|---|---|---|---|---|
| $\alpha$-WGAN | Low | High | H | 0.496 | 15.676 | 0.77+-0.03 | 0.86+-0.01 | 0.68+-0.03 |
| VQ-VAE Adap | Low | High | H | 0.948 | 10.605 | 0.83+-0.01 | 0.85+-0.02 | 0.77+-0.03 |
| VQ-VAE Adap | High | High | H | 0.991 | **6.655** | 0.84+-0.05 | **0.92+-0.02** | **0.79+-0.08** |
| VQ-VAE Baur | Low | High | H | 0.947 | 10.982 | 0.82+-0.04 | 0.82+-0.05 | 0.74+-0.04 |
| VQ-VAE Baur | High | High | H | **0.998** | 6.737 | **0.85+-0.05** | 0.90+-0.03 | 0.75+-0.09 |
| $\alpha$-WGAN | Low | Low | H | 0.505 | 12.281 | 0.78+-0.03 | 0.86+-0.01 | 0.65+-0.01 |
| VQ-VAE Adap | High | Low | H | 0.951 | 7.255 | 0.78+-0.03 | **0.88+-0.02** | 0.71+-0.04 |
| VQ-VAE Adap | Low | Low | H | **0.994** | **5.055** | **0.84+-0.02** | 0.87+-0.01 | **0.78+-0.02** |
| VQ-VAE Baur | High | Low | H | 0.953 | 7.227 | 0.78+-0.03 | 0.86+-0.03 | 0.71+-0.04 |
| VQ-VAE Baur | Low | Low | H | 0.991 | 6.538 | 0.83+-0.05 | 0.86+-0.06 | 0.75+-0.03 |
| $\alpha$-WGAN | Low | High | P | 0.510 | 15.801 | 0.76+-0.02 | 0.86+-0.01 | 0.73+-0.03 |
| VQ-VAE Adap | Low | High | P | 0.895 | 10.982 | 0.81+-0.04 | 0.87+-0.03 | 0.80+-0.03 |
| VQ-VAE Adap | High | High | P | 0.981 | **7.346** | 0.86+-0.05 | **0.94+-0.02** | **0.86+-0.05** |
| VQ-VAE Baur | Low | High | P | 0.892 | 11.362 | 0.80+-0.03 | 0.84+-0.03 | 0.77+-0.05 |
| VQ-VAE Baur | High | High | P | **0.995** | 7.508 | **0.88+-0.07** | 0.91+-0.06 | 0.81+-0.10 |
| $\alpha$-WGAN | Low | Low | P | 0.545 | 12.351 | 0.76+-0.02 | 0.87+-0.01 | 0.72+-0.01 |
| VQ-VAE Adap | High | Low | P | 0.939 | 7.662 | 0.78+-0.03 | 0.87+-0.02 | 0.73+-0.03 |
| VQ-VAE Adap | Low | Low | P | **0.993** | **5.566** | **0.83+-0.02** | **0.88+-0.02** | **0.78+-0.02** |
| VQ-VAE Baur | High | Low | P | 0.938 | 7.664 | 0.73+-0.05 | 0.83+-0.06 | 0.70+-0.07 |
| VQ-VAE Baur | Low | Low | P | 0.990 | 6.834 | 0.82+-0.03 | 0.87+-0.03 | 0.77+-0.04 |
| $\alpha$-WGAN | Low | High | PB | 0.511 | 15.807 | 0.75+-0.02 | 0.85+-0.01 | 0.73+-0.03 |
| VQ-VAE Adap | High | High | PB | 0.984 | **7.573** | **0.88+-0.05** | **0.93+-0.02** | **0.84+-0.06** |
| VQ-VAE Baur | High | High | PB | **0.993** | 10.818 | **0.88+-0.05** | 0.92+-0.07 | 0.82+-0.09 |
| $\alpha$-WGAN | Low | Low | PB | 0.542 | 12.361 | 0.75+-0.02 | **0.86+-0.01** | **0.72+-0.02** |
| VQ-VAE Adap | High | Low | PB | 0.941 | **7.628** | **0.76+-0.02** | 0.84+-0.08 | 0.71+-0.05 |
| VQ-VAE Baur | High | Low | PB | **0.943** | 7.966 | 0.74+-0.03 | 0.84+-0.02 | 0.71+-0.08 |

Table 2: Table showcasing the results of all trained models and their upsampled/downsampled results. We are showing Maximum Mean Discrepancy on log scale for ease of visualization. For resolution *Low* means *64x64x64* and *High* is *192x256x192*. For training mode *H* is *control subjects*, *P* is a model *fine-tuned on pathological dataset* after being trained on H and *PB* is a model *trained from scratch only on pathological dataset*.

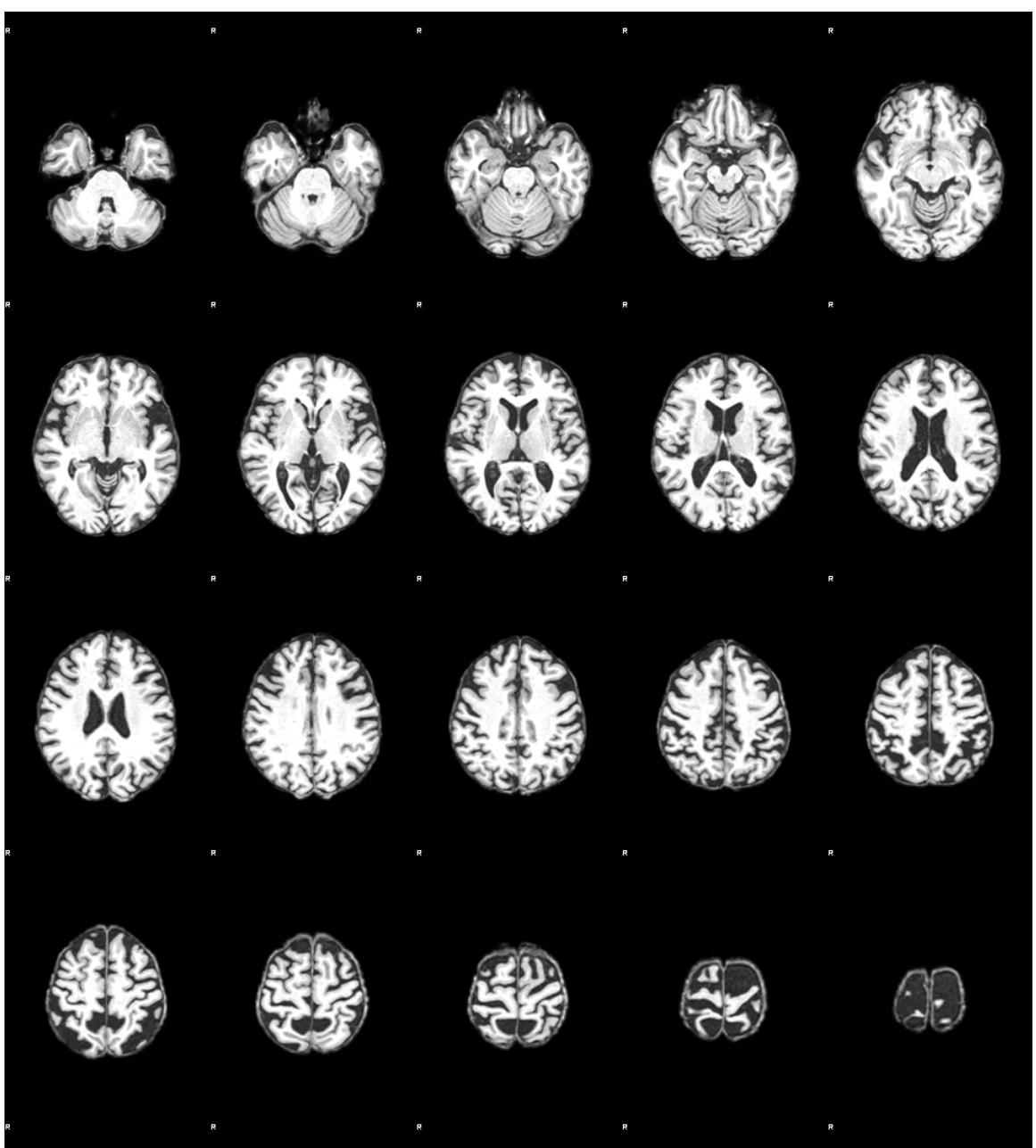

Figure 5: Axial slice based representation of the an original image

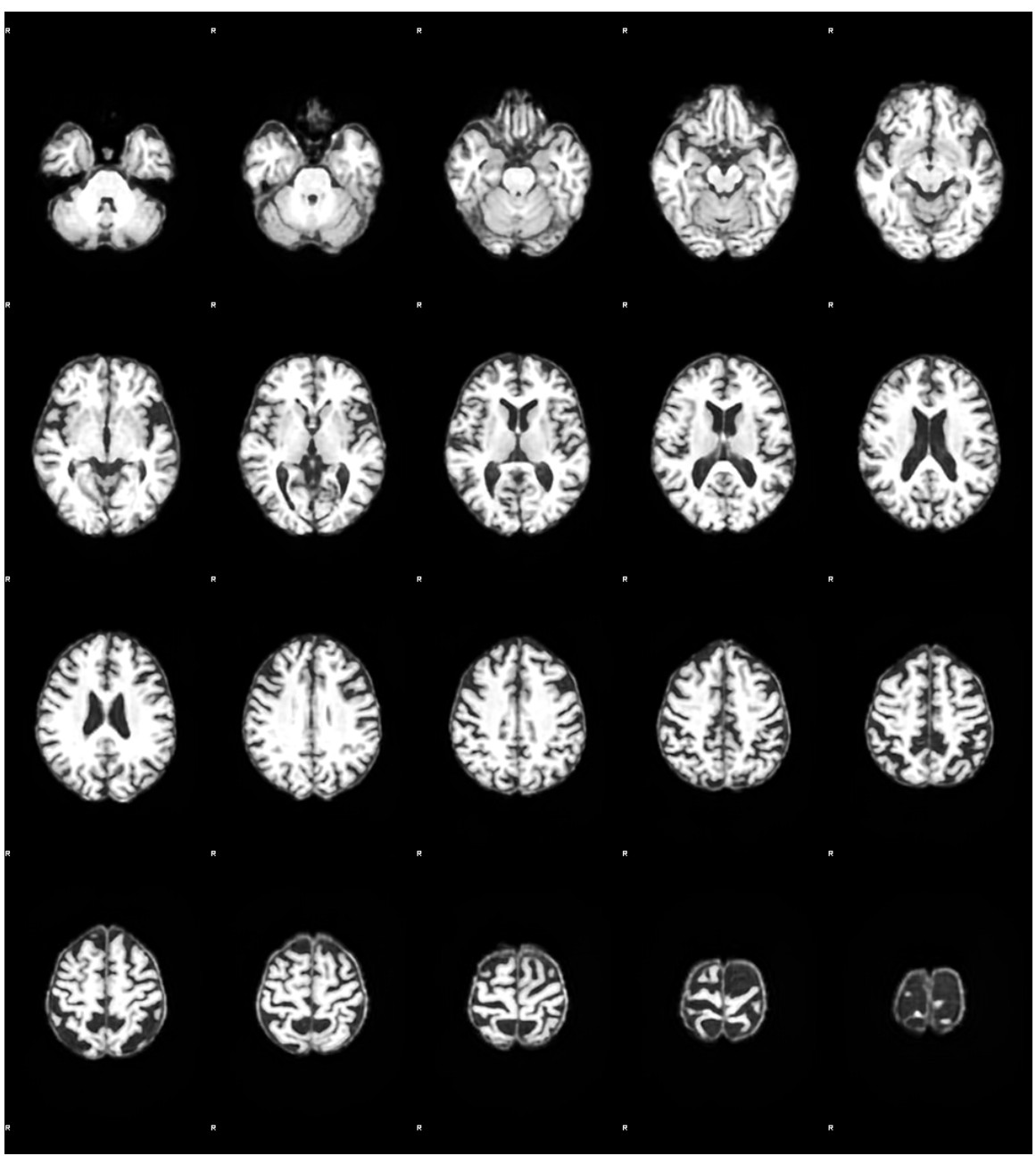

Figure 6: Axial slice based representation of the reconstruction of Figure 5

