# OpenReview forum: "Neuromorphologicaly-preserving Volumetric data encoding using VQ-VAE"
_MIDL.io/2020/Conference — Submitted to MIDL 2020_

### Official Review · AnonReviewer3 · 2020-03-08
**Modified VQ-VAE for encoding full-resolution 3D brain MRI.**

**Rating:** 2
**Confidence:** 4

**Summary:**

The paper modifies VQ-VAE for encoding a full-resolution 3D brain MRI to a small percentage of the initial size while preserving morphology. The modification includes replacing the 2D convolution blocks with 3D blocks and using recently proposed initialization and loss functions. The proposed idea is evaluated on the standard dataset, and the presented results demonstrate improved performance in terms of image consistency metrics.

**Strengths:**

Adaptation of widely popular VQ-VAE into 3D setup is one of the biggest strengths of this paper. This extension, especially in the medical imaging domain, could be significant as working with 3D data has been one of the critical challenges of this field. Having a high-fidelity 3D model should also encourage learning and analyzing latent space for these applications.

**Weaknesses:**

The proposed methodology seems to be a holistic approach to integrating useful properties from previous works in one place. In this sort of methodological contribution, it is natural to expect ablation studies that analyze the effect of such inclusion. Not just ablation study, even proper discussion about some inclusion seems missing. For instance, it is not clear why the loss function (Baur et al., 2019) was used.

Also, the paper is poorly written, with claims not backed by the proper references and numerous errors. For example, there are claims like modeling "gradients instead of pixel intensities which almost always work better" without any reference.

**Detailed Comments:**

Given the scope in improving the reconstruction image fidelity and image structure, this paper could be a critical contribution. However, the paper can be much improved in terms of writing. Some of the minor suggestions for improvements are listed as:
- While describing model architecture, p(z) is referred to as posterior instead of prior
- Voxel-Based Morphometry (VBM) analysis is considered to demonstrate the preservation of morphology of the decoded samples. It would be better if the authors cite the appropriate prior works proposing VBM analysis or briefly explain the approach as it would enhance the readability of the paper.
- Fig 1, is not readable in the print version. Authors are encouraged to use a better approach.
- In the discussion section, the 'dice' score is misspelled as 'die' score.
- Finally, in the title, the term 'Neuromorphologicaly' seems to be misspelled. If not, can the authors explain more about the term used?

**Justification Of Rating:**

Although paper aims towards solving a significant problem in medical imaging analysis, the motivation and the experiments are not aligning. As explained earlier, the ablation study is critical when the proposed methodology is presented as a combination of prior works. Furthermore, the paper is not well written, and the usage of some approaches are not clearly explained.

**Paper Type:**

methodological development

**Questions To Address In The Rebuttal:**

- What is the motivation behind considering Baur et al., 2019, in this setup?
- Why ICNR initialization is used, and what happens if such initialization is not used? Can the authors include some ablation studies on this type of critical inclusion?


**Special Issue:**

no

---

> ### Author Response · Authors · 2020-03-27
> **Response to AnonReviewer3**
>
> In regards to the ablation studies, we would like to bring to the reviewer's attention that each network was trained for 7 days and the finetuned ones were further trained for 4 days, resulting in a total of 1824 hours of GPU compute time. If we were to run a single ablation study with the same experimental framework it would double the compute type which would be infeasible with our current equipment. In regards to the SubPixel Convolution's ICNR initialization, we would like to redirect the reviewer towards the Appendix section of [1] where is show how the ICNR impacts the SubPixel Convolution and to [2] for why we want our Upsampling Convolution to be close to a Nearest-Neighbor Convolution.
>
> Our main aim was to compare against the state of the art [3] and show that the proposed network brings improved fidelity and morphological preservation. Then we further proposed an improved loss function that brought increased fidelity and morphological preservation. Additionally, it also showcases that the network structure and training dynamics are not dependent on the choice of loss function.
>
> In regards to the pixel intensities statement, due to the page limit, we have suppressed the [4] citation where it is shown that the L2 norm of the residuals underperforms compared with the L2 norm of the image gradients. We will add the additional reference in the final version and also tone down the statement.
>
> Voxel-Based Morphometry (VBM) is considered the defacto tool for neuromorphometrical analysis as seen by the track record of [5]. VBM provides a spatially varying univariate statistical map of morphological preservation, which is one of the main properties we want to demonstrate our reconstruction have. On top of that, we are also presenting results for dice scores as a proxy metric of the morphological preservation. We have not expanded this point due to the page limit, but we will do our best to include some details in the final version of the paper.
>
> As we have already stated, we will do our best to further improve the readability and content, and fix any typos that are left.
>
> [1] Aitken, A., Ledig, C., Theis, L., Caballero, J., Wang, Z. and Shi, W., 2017. Checkerboard artifact free sub-pixel convolution: A note on sub-pixel convolution, resize convolution and convolution resize. arXiv preprint arXiv:1707.02937.
> [2] Odena, A., Dumoulin, V. and Olah, C., 2016. Deconvolution and checkerboard artifacts. Distill, 1(10), p.e3.
> [3] Kwon, G., Han, C. and Kim, D.S., 2019, October. Generation of 3D brain MRI using auto-encoding generative adversarial networks. In International Conference on Medical Image Computing and Computer-Assisted Intervention (pp. 118-126). Springer, Cham.
> [4] Nal Kalchbrenner and Ivo Danihelka and Alex Graves 2016. Grid Long Short-Term Memory. In 4th International Conference on Learning Representations, ICLR 2016, San Juan, Puerto Rico, May 2-4, 2016, Conference Track Proceedings.
> [5] Ashburner, J. and Friston, K.J., 2000. Voxel-based morphometry—the methods. Neuroimage, 11(6), pp.805-821.

---

### Official Review · AnonReviewer4 · 2020-03-10
**How is it useful in medical imaging?**

**Rating:** 2
**Confidence:** 4

**Summary:**

The paper extends VQ-VAE architecture to 3D images and then shows that this VQ_VAE can yield good reconstruction (including neuromorphology preservation). The reconstruction is compared to another paper based on VAE + GAN architecture. Experimental section has both qualitative and quantitative comparison of the reconstruction with the VAE+GAN baseline.

**Strengths:**

Looking at the Dice GM, WM and CSF, the VQ-VAE clearly does better job than the baseline method, $\alpha$-WGAN (Kwon et al.). Similar results can be seen in Fig 2 , 3 and 4. It is good to know that VQ-VAE can be used to compress, although the application of that should be well motivated.

**Weaknesses:**

The paper has limited novelty. The idea of VQ-VAE including how to compute KL divergence with the prior and how to update the latent codes etc are used from the original VQ-VAE paper.  Using 3D convolution and U-net type skip connections are the architectural choices that seem new in this paper.

I have major concern with the applicability of this paper in the medical imaging context. Where do we use this? It is well known that VAE type architectures encode information in the latent space and usually compress while doing so. This paper shows that we can do similar thing by using VQ-VAE in the brain images. But, the applicability of their work in medical imaging is not well motivated. Can't we use numerous other compression algorithms in computer vision to compress the data? Why spend so much resource in training deep network to obtain data compression?



**Detailed Comments:**



Another question I have is with the latent codes. From the Fig. 1, it seems like the quantization and latent codes are present at different resolution (48^3, 12^3, 3^3). If we are to consider this as a generative model (which VAE is), how would we sample? and how would we interpret the latent codes at different depths? This is also related to the comparison with the baseline $\alpha$-WGAN, where the sampling is straightforward.

Similarly, since there are latent codes at different levels, how do we compute the latent dimension ?  48^3+12^3+3^3? If that is correct, the latent dimension in this case is much bigger compared to 1000 in the baseline.



**Justification Of Rating:**

I have major concerns with the novelty  and the relevance of the VQ-VAE in the medical imaging in general and brain image compression task in specific. In addition to that, other  concerns have been explained in the Detailed comments section.

**Paper Type:**

validation/application paper

**Special Issue:**

no

---

> ### Author Response · Authors · 2020-03-27
> **Response to AnonReviewer4**
>
> Our network is based on the VQ-VAE 2 [1] and the alteration might appear to be trivial compared to the original work, but the network design required substantial experimentation and extensive thought process to make it work stably in 3D. It should be brought to the attention that 3D networks are notorious to train efficiently and all the modifications we made were aimed at increasing the convergence speed and stability of the proposed network. We will further highlight and specify their contribution in the final paper.
>
> As per the paper we submitted, we show that the proposed model can preserve morphology and has high fidelity reconstructions. This finding enables a series of applications such as outlier detection, sample synthesis, decoupling sources of variability, etc. Furthermore, showing we obtained neuromorphologically preserving reconstructions allows us to head towards privacy-preserving data-sharing through the means of generative modeling since one needs to have morphologically correct samples to run any study on them. Those applications are not part of this paper as we have focused primarily on demonstrating the neuromorphologically preserving nature of the architecture, something which was never shown before.
>
> All generative models compress the data. However, most compression algorithms cannot be made generative. The proposed method can do both, be generative and compress information. Furthermore, compressibility in a network that can be made generative means that we achieve a compact data representation simplifying future applications in generative modeling and code space disentanglement.
>
> This paper does not sample from the VQ-VAE. It only demonstrates the ability to reconstruct neuromorphologically preserving images. Sampling can be done through the use of PixelCNNs as per [1]. We are currently working on enabling it by extending the PixelCNN++ [2] to 3D in a similar fashion with [3] and extend the comparison in the next publication.
>
> We have not provided any visualization of the different depth codes but the original VQ-VAE 2 paper [1] shows that different depths represent different frequencies of the generated image.
>
> Indeed the latent representation is bigger than the baseline however AutoEncoder/Generative Adversarial Network based models have an intrinsically small latent space since they do not represent it hierarchically. Because of this if we were to increase the latent space dimensionality of the competing method they would run out of memory or become unstable during training, while the proposed method does not.
>
> In the final version, we will further improve the readability, highlight the possible usages of our method, and fix any typos that are left.
>
>
> [1] Razavi, A., van den Oord, A. and Vinyals, O., 2019. Generating diverse high-fidelity images with vq-vae-2. In Advances in Neural Information Processing Systems (pp. 14837-14847).
> [2] Tim Salimans and Andrej Karpathy and Xi Chen and Diederik P. Kingma 2017. PixelCNN++: Improving the PixelCNN with Discretized Logistic Mixture Likelihood and Other Modifications. In 5th International Conference on Learning Representations, ICLR 2017, Toulon, France, April 24-26, 2017, Conference Track Proceedings. OpenReview.net.
> [3]Pombo, G., Gray, R., Varsavsky, T., Ashburner, J. and Nachev, P., 2019, October. Bayesian Volumetric Autoregressive generative models for better semisupervised learning. In International Conference on Medical Image Computing and Computer-Assisted Intervention (pp. 429-437). Springer, Cham.

---

### Official Review · AnonReviewer2 · 2020-03-12
**An adaptation of the VQ-VAE for 3D (medical) data with an adaptive reconstruction loss function, facilitating high reconstruction fidelity for complex 3D data**

**Rating:** 3
**Confidence:** 4
**Recommendation:** Poster

**Summary:**

The paper describes an adaptation of the VQ-VAE for 3D (medical) data with an adaptive reconstruction loss function, ultimately facilitating high reconstruction fidelity for complex 3D data. To date, this has been very challenging due to limitations of computational resources and is thus highly relevant. In fact, this holds potential to enable a plethora of future research in the field of 3D medical image analysis.

**Strengths:**

- Clearly motivated and well written
- Trying to solve a relevant problem: high fidelity reconstruction using VQ-VAEs in 3D data
- This opens up many opportunities for tasks such as unsupervised anomaly detection
- Testing for statistical significance is highly appreciated
- Very interesting loss formulation, which can surely be useful in many other settings as well

**Weaknesses:**

- I feel the methodology could be expanded to discuss multiple design choices and formulations in greater detail
- For instance, the proposed VQ-VAE model has something that I would refer to as "skip-connections", i.e. modeling / compressing features at different scales. In this context, I believe a comparison to a normal VAE (or Autoencoder) which has such connections / compresses at multiple scales would be required and show the real benefit of such VQ.
- Generally, I feel the variational part of the method was hardly addressed. I believe that exemplary image synthesis using the VQ-VAE (it is supposed to be a generative model after all) or residual-based anomaly detection would have strengthened the paper a lot
- I suspect some details on the loss function from Baur et al. are missing: I would assume that the weighting of the different loss components plays an important role. How were the different terms weighted during optimization?
- The baseline the authors compare against operates at a very different resolution as the VQ-VAE does; In this context, I am not sure whether the provided metrics are all valid, e.g. the Dice-score might be generally lower at lower resolutions. Can the authors please elaborate more on this validity?

**Detailed Comments:**

Minor:
- In the first sentence of the methodology, you refer to p(z) as posterior, though it is the prior
- Section 4.2: "Gray Mater"
- Discussion section: "die scores"

**Justification Of Rating:**

I consider this work an enabler for future research on 3D medical image analysis in various directions. In addition, the paper is well written. Thus, the work deserves to be presented to a wider audience, if experimental design choices can be elaborated more and additional baselines can be added (if appropriate).

**Paper Type:**

both

**Questions To Address In The Rebuttal:**

If fidelity is of such great importance, why not use an adversarial network on top of the VQ-VAE to "learn" a loss function? Would computational constraints still allow to formulate discriminator?

**Special Issue:**

no

---

> ### Author Response · Authors · 2020-03-27
> **Response to AnonReviewer2**
>
> During the paper's development, we have also used AutoEncoder and Variational AutoEncoders, but there were a few issues with them. With AE we do not have an efficient principled way of sampling from them. One, in theory, could do exactly what the authors of VQ-VAE do but that is a methodological paper of its own. As for VAE, it is already known that they suffer from blurry reconstructions. This coupled with the preprocessing needed for the VBM will render the brain reconstructions useless for further analysis. Furthermore, it would distract from the main point of the paper, which is that we can obtain neuromorphologically correct reconstructions.
>
> As stated in the above paragraph the aim was obtaining neuromorphologically correct reconstructions, thus the sampling from the quantized space to enable generative behavior was out of the remit of this paper. We are currently working on enabling it by extending the PixelCNN++ [1] to 3D in a similar fashion with [2]. Furthermore, applications of the proposed network, such as anomaly detection and unsupervised segmentation are out of the remit of our work and would further dilute the message we tried to convey.
>
> Baur loss' training regime was determined on the training set. The details will be added in the final version of the paper but we will write them here as well. The loss' components weighting was done as follows, the L1 and L2 losses of the residuals have a weight of 1 throughout the training, the image gradients have a weight of 0 for the first 100000 batches (realistically this number can be changed as long as it is after the threshold where the gray matter can be seen ever so slightly) and is increased linearly until it reaches 1 over 10000 batches, then it keeps increasing at the same rate (linear) until it reaches a maximum value of 5 where it plateaus. This regime was not varied at all across our trained models.
>
> We have trained our proposed model on both 192x256x192 as well as 64x64x64, we will make it more clear in the final version. We ask the reviewer to look into the appendix in Table 2 where all the experiments that we have done are outlined. Even on the same spatial resolution, we are obtaining better results than the baseline as presented in the extended table. We chose to highlight the 192x256x192 models purely because of their practical implications in further research. Also, it should be noted that Generative Adversarial Networks are non-trivial to run at 192x256x192 due to memory constraints and even harder to balance training dynamics. Those were also the reasons that the baseline was not trained at full resolution.
>
> A discriminator network is just as good as the dataset it was trained on and the programmed dynamics might not generalize to another dataset. Furthermore, adversarial networks are notoriously known to hallucinate textures as seen in our baseline and introduce instability during training. All of the above, combined with the engineering overhead required to finetune the dynamics between the VQ-VAE and Discriminator networks, made us chose not to go down that route.
>
> In the final version we will further improve the readability, better highlight the full array of experiments that have been done, and fix any typos that are left.
>
> [1] Tim Salimans and Andrej Karpathy and Xi Chen and Diederik P. Kingma 2017. PixelCNN++: Improving the PixelCNN with Discretized Logistic Mixture Likelihood and Other Modifications. In 5th International Conference on Learning Representations, ICLR 2017, Toulon, France, April 24-26, 2017, Conference Track Proceedings. OpenReview.net.
> [2]Pombo, G., Gray, R., Varsavsky, T., Ashburner, J. and Nachev, P., 2019, October. Bayesian Volumetric Autoregressive generative models for better semisupervised learning. In International Conference on Medical Image Computing and Computer-Assisted Intervention (pp. 429-437). Springer, Cham.

---

### Meta-Review · Area_Chair1 · 2020-04-07
**MetaReview of Paper70 by AreaChair1**

**Rating:** 2

**Metareview:**

There is agreement among the reviewers over both some positive aspects and some negative. I've carefully reviewed the material and concluded that the paper is not ready for publication.

All reviewers seem to agree that this is an interesting application of the VQVAE work to medical images (and I agree as well). The novelty itself is challenged by all reviewers, and one important (and likely important) answer from the authors is that there is work involved to scaling up VQ-VAE and making it stable -- and while this is true, this answers the challenges of the work, not the amount of novelty (it also seems weak to use runtime as an argument for why ablation studies cannot be done).

I think that a proper and thorough application paper is also appropriate for MIDL. Unfortunately, the reviewers challenge the scope and story of the paper as well, and I agree that there is significant confusion/contradiction, including the responses. For example, there is significant confusion about the generative aspect of the generative models, to which the authors say (R2, second para of response) that sampling from a generative model was beyond the scope of the paper, and what they focus on compression. But then, when asked to compare with compression mechanisms, they emphasize (R4, second para of response) that there is value in having a generative model vs just a compression algorithm. Surely if generative models are important they should show generative behaviour (e.g. sampling) or if compression is important than comparing with compression algorithms (even classical ones based on wavelets, etc) is warranted. Furthermore, the application to the medical domains is questioned (appropriately, I think), with the response being that they are important but beyond the scope of the paper. Individually, some of these answers are sensible, but together they highlight why the paper falls short of either a strong methodological paper or a strong application paper.

There are good ideas in this paper, and I really encourage the authors to continue. My main suggestion coming our from the review process is that first the story and positioning needs to be straightened out, and based on the that appropriate changes be made. For example, if methodology is emphasized, then more novelty seems to be necessary, and either sampling should be addressed as part of the generative model, or the generative claim/emphasis should be dropped (but the experimental tasks are appropriate). If the application is emphasized, then the method should be applied to more than the current applications, but maybe one that the authors stated as the downstream goal, such as anomaly detection. I hope the authors take the productive comments here and improve the paper, it would be great to have it published in a future conference.

**Paper Type:**

validation/application paper

**Special Issue:**

no

---

### Decision · Program_Chairs · 2020-04-11

Reject